

# Transcriptome analysis of Chinese mitten crabs (*Eriocheir sinensis*) gills in response to ammonia stress

Zhengfei Wang[1,*], Yue Wang[1,*], Yayun Guan[1], Zhuofan Chen[1], Yaotong Zhai[1], Ya Wu[2], Ying Zhou[1], Jinghao Hu[1] and Lulu Chen[1]

[1] Jiangsu Key Laboratory for Bioresources of Saline Soils, Jiangsu Synthetic Innovation Center for Coastal Bio-agriculture, Jiangsu Provincial Key Laboratory of Coastal Wetland Bioresources and Environmental Protection, School of Wetlands, Yancheng Teachers Universtiy, Yancheng, Jiangsu Province, China
[2] College of Life Sciences, Nanjing Normal University, Nanjing, Jiangsu Province, China
* These authors contributed equally to this work.

## ABSTRACT

The Chinese mitten crab (*Eriocheir sinensis*) is an important commercial species in China. *E. sinensis* is typically farmed in rice-crab symbiosis, as an important ecological farming model. However, *E. sinensis* is often exposed to a high ammonia environment due to the application of nitrogen fertilizers essential for rice growth. We investigated the molecular mechanisms in the gills of *E. sinensis* exposed to high ammonia at transcriptional and histological levels. We randomly assigned *E. sinensis* to two groups (control group, CG; ammonia stress group, AG), and gill samples were excised from the CG and AG groups for histopathological and transcriptome analyses. The histopathological evaluation revealed that ammonia stress damaged the gills of *E. sinensis*. The transcriptome analysis showed that some essential genes, including *Xanthine dehydrogenase* (*XDH*), *Ubiquitin C-terminal hydrolase-L3* (*UCHL3*), *O-linked N-acetylglucosamine transferase* (*OGT*), *Cathepsin B* (*CTSB*), and *Ubiquitin-conjugating enzyme E2 W* (*UBE2W*) changed significantly during ammonia exposure. These genes are related to ammonia detoxification, the immune response, and apoptosis. This study demonstrated the molecular response mechanism of *E. sinensis* gills to ammonia stress at the transcriptional and histological levels. This study provides insight for further study on the molecular mechanism of ammonia stress in crustaceans and supplies technical support for rice crab symbiosis.

# INTRODUCTION

*Eriocheir sinensis* is an important commercial species in China and a Chinese traditional aquatic treasure (*Chen, Zhang & Shrestha, 2006*). Rice-crab symbiosis has gradually become an effective farming model in the last century. Rice-crab symbiosis is the organic combination of rice and *E. sinensis*. The rice provides the crabs with an excellent living environment and diversified food to reduce the input of artificial feed. *E. sinensis*,

Corresponding authors
Zhengfei Wang,
wangzf01@yctu.edu.cn
Lulu Chen, chenll@yctu.edu.cn

accelerates the turnover of organic matter by feeding on aquatic animals and plants in the rice paddies and facilitates the uptake of nutrients by rice roots. *E. sinensis* continuously provides nourishment for the rice. Thus, the rice-crab symbiosis model improves resource utilization and rice yield and has overall benefits (*Yan et al., 2013*). Nitrogen is a key factor in rice growth and yield (*Tang et al., 2019*). However, applying nitrogen fertilizer in a rice-crab symbiotic system exposes *E. sinensis* to high levels of ammonia during the short term (*Yan et al., 2013*).

Ammonia is one of the most limiting factors in aquaculture, and is mainly produced during the mineralization of organic wastes, including uneaten feed and feces, and it is also derived from excretion by farm animals (*Sinha et al., 2014*; *Huang et al., 2021*; *Luke, Ravi & Colin, 2005*; *Romano & Zeng, 2007*). Ammonia exists in water as unionized ($NH_3$) and ionized ($NH_4^+$) forms. $NH_3$ is more toxic to crustaceans because it is lipid-soluble and easily penetrates the lipid bilayer (*Benli, Köksal & Özkul, 2008*). Many studies have reported that aquatic animals, including crustaceans, have excretory and detoxification mechanisms to resist ammonia stress (*Hong et al., 2007b*). For example, ammonia is excreted from the body through several transporters, such as the Rhesus (Rh) glycoprotein, $Na^+/K^+$-ATPase (NKA), and the $Na^+/H^+$-exchanger (NHE) (*Zhang et al., 2021*; *Shen et al., 2021*; *You et al., 2018*). Moreover, crustaceans convert ammonia to glutamine and urea through different pathways (*Hong et al., 2007b*; *Pan et al., 2018*; *Li et al., 2018*). Ammonia accumulates in crustaceans and causes serious damage if it reaches a certain threshold. Ammonia affects gas exchange, changes the histological structure of organs, and can cause death (*Diodato, Amin & Comoglio, 2019*; *Lu et al., 2017*; *Kır, Kumlu & Eroldoğan, 2004*; *Chen & Cheng, 1993*; *Romano & Zeng, 2013*). Ammonia also alters energy metabolism, damages immune capacity, causes oxidative stress, and induces apoptosis (*Cheng et al., 2019*; *Xu & Zheng, 2020*; *Tang et al., 2022*; *Liang et al., 2019*). Therefore, understanding the mechanisms and consequences of ammonia toxicity in *E. sinensis* is essential.

The gills of crustaceans play a crucial role in osmoregulation, the immune response, respiration, and toxicology. Additionally, crustaceans, as ammonia-producing animals, actively excrete ammonia in a gradient inward direction. The gills are thought to be the main excretory organ of ammonia in aquatic crustaceans (*Wang et al., 2021*; *Yue et al., 2010*). Many genes and pathways are related to defending against pathogens because the gills close contact with the water and are exposed to bacteria and toxins (*Shen et al., 2021*). Thus, the gills are the proper organ to investigate the response mechanism of ammonia exposure in crustaceans.

In this study, frozen sectioning and Illumina sequencing were performed to investigate the effects of high ammonia stress on the gills of adult *E. sinensis*. The results revealed the toxic effects of a high ammonia level on *E. sinensis*. Our results provide a better understanding of the molecular mechanism of gills in *E. sinensis* response to ammonia stress and provide new insight into aquaculture management.

## MATERIALS AND METHODS

Portions of this text were previously published as part of a preprint (https://doi.org/10.21203/rs.3.rs-2372193/v1).

## Experimental materials

Twenty healthy and active *E. sinensis* (120 ± 10 g) were collected from the Renmin Road market in Yancheng (Jiangsu, China). The experimental crabs were placed in tanks (30 × 18 × 20 cm) filled with 4 L of water (temperature: ~17 °C; pH: ~7.5) for 1 week of acclimatization. During the acclimatization period, the crabs were fed a commercial diet at 3% of body weight at noon every day. Water in the tank was replaced at 9:00 am daily and uneaten food was removed.

## Animals and the ammonia stress experiment

After acclimation, six active crabs were selected for the ammonia stress experiment and randomly divided into the control group (CG; clean water was added) and the experimental group (AG; 325.07 mg/L $NH_4Cl$) was added. Each group had three replicates (each replicate had one sample) and the crabs were held in three identical tanks. The ammonia concentration for the experimental group was determined according to the study (*Hong et al., 2007a*) and our pre-experiment. The 96 h-LC50 (median lethal concentration for experimental individuals within 96 h) was 325.07 mg/L $NH_4Cl$. So, the concentration of the experiment was set to 325.07 mg/L. The stress experiment lasted 24 h. Neither group of crabs molted during the experiment. After the stress experiment, three samples each from the control group (CG-1, CG-2, CG-3) and ammonia stress group (AG-1, AG-2, AG-3) were selected, and gill tissues were obtained and immediately frozen in liquid nitrogen for RNA extraction.

## RNA extraction, cDNA library construction, and sequencing

Total RNA of the gill tissues was extracted using TRIzol reagent (Aidlab, Beijing, China) following the manufacturer's procedure. The concentration and purity of the RNA were determined with the Nanodrop 2000 spectrophotometer, and RNA integrity was verified by 1% agarose gel electrophoresis. The calculation of the statistical power for our RNA-seq data was performed by RNA Seq Power Calculator (*Ching, Huang & Garmire, 2014*). The biological replicates used to achieve the claimed statistical power is three. RNA sequencing (RNA-seq) is one of the most widely used technologies in transcriptomics, and cDNA libraries were constructed using an Illumina TruseqTM RNA sample prep Kit (Illumina, San Diego, CA, USA). Finally, the libraries of all samples were sequenced on the Illumina Hiseq/Miseq (Illumina, San Diego, CA, USA).

## Transcriptome assembly and functional annotation

Before assembly, sequences containing poly-N reads (>10% reads) or low-quality reads (<20 bp reads, q-value <20) were removed using SeqPrep (https://github.com/jstjohn/SeqPrep) and Sickle (https://github.com/najoshi/sickle) to obtain the clean reads. Based on the existing reference genome, the clean reads were assembled using the Cufflinks software (https://cole-trapnell-lab.github.io/cufflinks/) (*Haas et al., 2013*). Transcripts without annotation information were compared with known transcripts, and potential new transcripts are functionally annotated (Table S1). The clean data (reads) after quality control was aligned to the reference genome for subsequent transcript assembly and

expression calculation. Sequence alignment analysis was performed using HiSat2 (https://daehwankimlab.github.io/hisat2/). The clean reads were also assembled *de novo* to form transcripts using Trinity software (*Trapnell et al., 2010*). Both reference-based and *de novo* assembly were conducted, and the results were basically identical (Fig. S1, Table S1 and S2). The subsequent analysis of this study was based on reference-based result. Subsequently, the genes were annotated using the Basic Local Alignment Search Tool X (Blast X) (version 2.2.25, E value < $1e^{-5}$) according to several databases, including the NCBI non-redundant protein sequence database (NR), Pfam database (https://pfam.sanger.ac.uk/), COG database (Clusters of Orthologous Groups of proteins, https://www.ncbi.nlm.nih.gov/COG/), GO database (Gene Ontology, https://www.geneontology.org), and the KEGG database (Kyoto Encyclopedia of Genes and Genomes, https://www.genome.jp/kegg/).

## Analysis of differentially expressed genes

The expression levels of the genes were calculated by reads per kilobase per million (RPKM). RPKM: Reads Per Kilobase of exon model per Million mapped reads. Represents the number of reads per million derived from a gene per kilobase length. RPKM is the number of reads mapped to the gene divided by the number of all reads mapped to the genome (in million) and the length of the RNA (in kb) (www.metagenomics.wiki). The calculation formula of RPKM was adopted as the previous study: $RPKM = 10^9 \times C/(N \times L)$, where $C$ represented the number of reads for a gene, $N$ represented the total number of reads, and $L$ represents the transcript length corresponding to the gene (*Mortazavi et al., 2008*). The read counts of each sequenced library were adjusted using edgeR v.3.24.3 software and the scale normalization factor before analyzing the DEGs. edgeR v.3.24.3 is a widely used software that allows us to estimate the recounts of each gene. The differential expression analysis is being conducted at the gene level. Genes were considered DEGs when the false discovery rate (FDR) was <0.05 and $\log_2$|fold change (FC)| was ≥1 as analyzed with the edgeR statistical package (http://bioconductor.org/packages/stats/bioc/edgeR/). All DEGs were mapped to different categories using GO functional and KEGG pathway enrichment analyses to understand the functions and biological pathways of the DEGs. GO functional enrichment analyses were conducted using Goatools (https://github.com/tanghaibao/Goatools), and the q-value was set to 0.05. The statistical enrichment of candidate genes in the KEGG pathways was analyzed using KOBAS (http://kobas.cbi.pku.edu.cn/). Pathways with FDR < 0.05 were considered significantly enriched.

## Ammonia histopathological test

Twelve healthy crabs were selected and randomly divided into three groups (four crabs per group). The three groups of crabs were placed in three identical tanks. One group was the control group and the other two groups were experimental groups (received 325.07 mg/L $NH_4Cl$). The stress times for the two experimental groups were 24 and 48 h respectively. After the 24 and 48 h ammonia exposure experiment, the gills were collected and stored at −20 °C for 1 h. Then, the gills were cut into 5 µm slices with the Leica CM1950 Freezing slicer (Lecia Biosystems GmbH, Nussloch, Germany). The slices were fixed in 10%

formaldehyde for 30–60 s and rinsed in water for 1–2 s. The slices were immersed in dye containing hematoxylin for 3–5 min. Next, the slices were washed in water for 5–10 s to remove the hematoxylin. The slices were immersed in 1% hydrochloric acid/alcohol for 2 s and rinsed in water for 1–2 s. Finally, the slices were immersed in a dyeing tank containing eosin for 30–60 s, and the surface was rinsed with clean water. Finally, the slices were observed under an optical microscope (LEICA DM4000B; Leica, Wetzlar, Germany).

## Quantitative real-time polymerase chain reaction

Five genes were selected randomly for quantification using real-time PCR (qRT-PCR) to verify the accuracy of the sequencing data. The RNA of the gills used for qRT-PCR was isolated from the crabs used for Illumina sequencing. Primer Premier 5 software was used to design the specific primers of the five selected DEGs (Table 1). *β-actin* was the reference gene to normalize the expression levels of the target genes. The qRT-PCR analysis was performed on an Applied Biosystems 7500 real-time PCR system (Applied Biosystems, Thermo Fisher Scientific, Waltham, MA, USA) using 2× SYBR Green qPCR Mix (Aidlab, Beijing, China). The PCR program was in four steps of 95 °C for 3 min, 40 cycles at 95 °C for 15 s, 60 °C for 15 s, and 72 °C for 25 s. Five samples and the internal reference gene were run three times and only one product was analyzed with the melting curve.
The relative expression levels of the genes were measured by the $2^{-\Delta\Delta CT}$ method (*Livak & Schmittgen, 2001*).

# RESULTS

## Histopathological evaluation

The gill filaments of *E. sinensis* were arranged neatly in the control group, and the gill cavity was not expanded or fractured (Fig. 1A). After 12 h of ammonia stress, the gill filaments remained neatly arranged and the gill cavity was partly expanded (Fig. 1B). The gill filaments were disordered after 24 h of ammonia stress and most of the gill cavity was expanded (Fig. 1C). In general, gill injury became more severe as ammonia exposure time was extended.

## Transcriptome sequencing and assembly

After RNA sequencing, the total number of raw reads generated from the control and ammonia stress groups was identified. The statistical power of this experimental design, calculated in RNA Seq Power is 0.76. The sequencing results were submitted to the NCBI Short Read Archive (SRA) database with accession numbers SRR22094391, SRR22094390, SRR22094389, SRR22094388, and SRR22094387, respectively. Due to the poor quality of CG-3 which was detected as an outlier sample by PCA analysis, it was removed and it was difficult to obtain samples for supplement in this season, there were therefore only two sets of CG. The elimination of the outlier sample might help us underly more gene regulatory mechanisms, more critically, the outlier sample which was belong to control group did not affect the data analysis practically. Through PCA analysis, the similarity between the samples between the control and experimental groups can be intuitively seen (Fig. S2). After removing the poly-N reads and low-quality reads, the total number of clean reads

**Table 1 Quantitative real-time PCR primers used in this study.**

| Gene name | Forward primer sequence (5′-3′) | Reverse primer sequence (5′-3′) |
|---|---|---|
| *β-actin* | GATGGTGGGAATGGGTCA | CCAACCGTGAGAAGATGACT |
| *CTSB* | CGGGCTCAGCGTTCAAGT | CGTAAGGGACGGAGTAGGAGT |
| *ALOX5* | GGACGGTGTTACAGTAGGGC | ACGCTCGTTGTCGTTGGT |
| *NRD1* | AGGAGCGGAACCTCACTG | CCTCGGGTAACGCCATTT |
| *PIK3CA* | AGACAGGCACAACAGCAACA | ACCAGCAGGAAGTCCGAGGC |
| *SLC5A8* | CCTTCAAGCCTCTATCAGTC | GTGCTCAGGAACCCAACG |

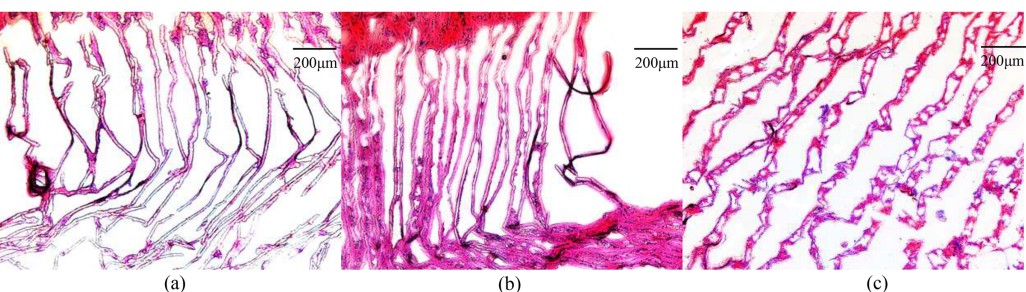

**Figure 1 Microscopic images of ammonia stress in gills from *E. sinensis*.** (A) Control group (B) 12 h after ammonia exposure (C) 24 h after ammonia exposure.

was obtained (Table 2) and the basic statistics of RNA-seq raw reads (Table 3). Subsequently, after reference-based assembly using the Cufflinks software, we obtained 69,555 sequences were more than 1,800 bp in length (Fig. S3).

## Functional annotation and classification

All the genes were annotated using the Basic Local Alignment Search Tool X (Blast X) according to several databases, including Blast X-NR, COG, Swissprot, GO, and KEGG (Table S1). Furthermore, the unigenes and transcript annotation results based on the *de novo* transcriptome were also provided (Table S2).

A total of 13,705 genes were identified in the GO database, which included the biological process (BP) category, cellular component (CC) category, and molecular function (MF) category. As results, the organonitrogen compound metabolic process (8.92%) was mostly annotated in the BP category. The term that dominated the CC category was cellular component (64.14%). Transporter activity (7.11%) and transmembrane transporter activity (6.8%) were mostly annotated in the MF category. A total of 12,324 genes were matched in the KEGG database. These genes were mapped to 192 pathways, and the top three of the most annotated pathways were ribosome biogenesis in eukaryotes (4.75%), pathways of neurodegeneration-multiple diseases (3.75%), and energy metabolism (3.25%).

**Table 2 The basic statistics of RNA-seq reads in *E. sinensis*.**

| Sample | Raw reads | Clean reads | Error% | Q20% | Q30% | GC% |
|---|---|---|---|---|---|---|
| AG-1 | 52,050,876 | 51,989,488 | 0.0274 | 96.71 | 92.42 | 48.11 |
| AG-2 | 55,844,004 | 55,772,056 | 0.0277 | 96.6 | 92.11 | 47.09 |
| AG-3 | 53,571,796 | 53,521,144 | 0.0272 | 96.83 | 92.55 | 47.71 |
| CG-1 | 61,916,296 | 61,844,070 | 0.028 | 96.45 | 91.89 | 47.7 |
| CG-2 | 54,608,990 | 54,547,802 | 0.0265 | 97.17 | 92.99 | 45.59 |
| CG-3 | 51,177,552 | 51,124,438 | 0.0265 | 97.22 | 92.98 | 45.36 |

**Table 3 The basic statistics of RNA-seq raw reads in *E. sinensis*.**

| Sample | NBCI SRA accession IDs | Number of bases (bp) | Error% | Q20% | Q30% | GC% |
|---|---|---|---|---|---|---|
| AG-1 | SRR22094391 | 5,451,191,808 | 0.0276 | 96.6 | 92.3 | 48.7 |
| AG-2 | SRR22094390 | 5,407,519,890 | 0.0273 | 96.7 | 92.4 | 45.8 |
| AG-3 | SRR22094389 | 5,353,043,620 | 0.0272 | 96.8 | 92.5 | 47 |
| CG-1 | SRR22094388 | 5,633,164,324 | 0.028 | 96.4 | 91.9 | 48.6 |
| CG-2 | SRR22094387 | 6,212,118,558 | 0.0282 | 96.4 | 91.8 | 49.7 |

## Identification and enrichment analysis of the DEGs

After setting the standard of the FDR < 0.05 and $\log_2|FC| \geq 1$ among the annotated genes, 254 DEGs (125 upregulated and 129 downregulated) were detected when *E. sinensis* was exposed to a high ammonia concentration (325.07 mg/L $NH_4Cl$) (Fig. 2). More genes were downregulated than upregulated. The DEGs were analyzed through GO and KEGG pathway functional enrichment to further investigate the physiological mechanism of the high ammonia concentration in *E. sinensis* gills (Figs. 3 and 4).

Among the significantly enriched GO terms ($P < 0.05$), some GO terms were related to ammonia excretion and the immune response, such as ion transport (GO:0006811), inorganic ion transmembrane transport (GO:0098660), anion transmembrane transport (GO:0098656), membrane protein complex (GO:0098796), endopeptidase regulator activity (GO:0061135), endopeptidase inhibitor activity (GO:0004866) and peptidase inhibitor activity (GO:0030414). In the KEGG analysis, some immune-related and apoptosis-related pathways were enriched, such as Phagosome (ko04145), Lysosome (ko04142), T cell receptor signaling pathway (ko04660), and Apoptosis (ko04210). Most of the DEGs were downregulated in these pathways. Many genes (*XDH*, *UCHL3*, *OGT*, *CTSB*, and *UBE2W*) related to ammonia, the immune response, and apoptosis were altered.

## Verification of the transcriptome data by qRT-PCR

To determine the accuracy of the sequencing data, we randomly selected five DEGs (*CTSB*, *ALOX5*, *NRD1*, *SLC5A8*, and *PICK3CA*) for qRT-PCR (Fig. 5, Data S1), with *β-actin* as the reference gene. The qRT-PCR results were calculated to access the fold-change in each gene, which was compared with the RNA-seq results. Figure 5 indicates that the qRT-PCR

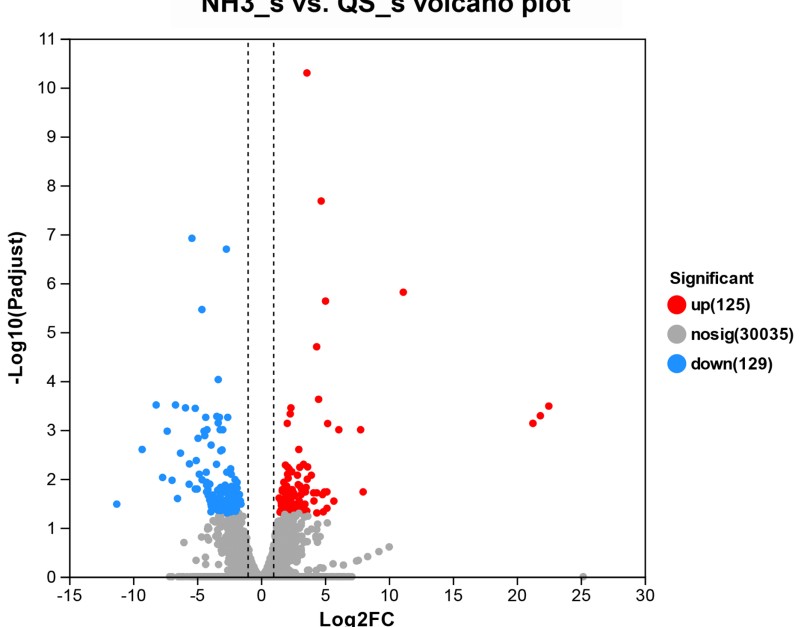

**Figure 2 Volcano plot of the differentially expressed genes in AG *vs* CG.** The blue scatters indicate the down-regulated genes, and the red scatters indicate the up-regulated genes. NH3_S represent ammonia stress group (AG) and QS_S represent control group (CG).

results were consistent with the sequencing results, demonstrating the reliability of the RNA-Seq and Trinity reference assembly.

# DISCUSSION

The concentration of ammonia in the water increases after applying nitrogenous fertilizer, which adversely affects the health status of *E. sinensis* in a rice-crab symbiotic system (*Rothuis, 1998*). Moreover, the gills of *E. sinensis* play an essential role in osmoregulation, immunity, and ammonia excretion (*Shen et al., 2021*). Therefore, we selected the gills as the research object to study the response of *E. sinensis* to ammonia stress. We analyzed gill histopathology, which revealed that tissue damage was more serious as stress duration was extended, and identified the DEGs related to ammonia detoxification, the immune response, and apoptosis based on the transcriptome data.

## Histopathological evaluation

The gill filament is the basic functional unit of the gill, and is responsible for excreting ammonia (*Weihrauch et al., 2002*). When the crabs were exposed to high ammonia (325.07 mg/L NH$_4$Cl) for 12 h, part of the gill cavity began to expand. When the gill tissues were exposed to the high ammonia level for 24 h, most of the gill cavities began to expand and some of the gill filaments ruptured. In contrast, the gill tissue was normal and no abnormalities were observed in the control group. Therefore, we inferred that the high ammonia stress caused damage to the gill tissues of *E. sinensis*, and that the damage was more severe the longer the stress time. These histological results indicate that ammonia damaged the gills at the cellular level.

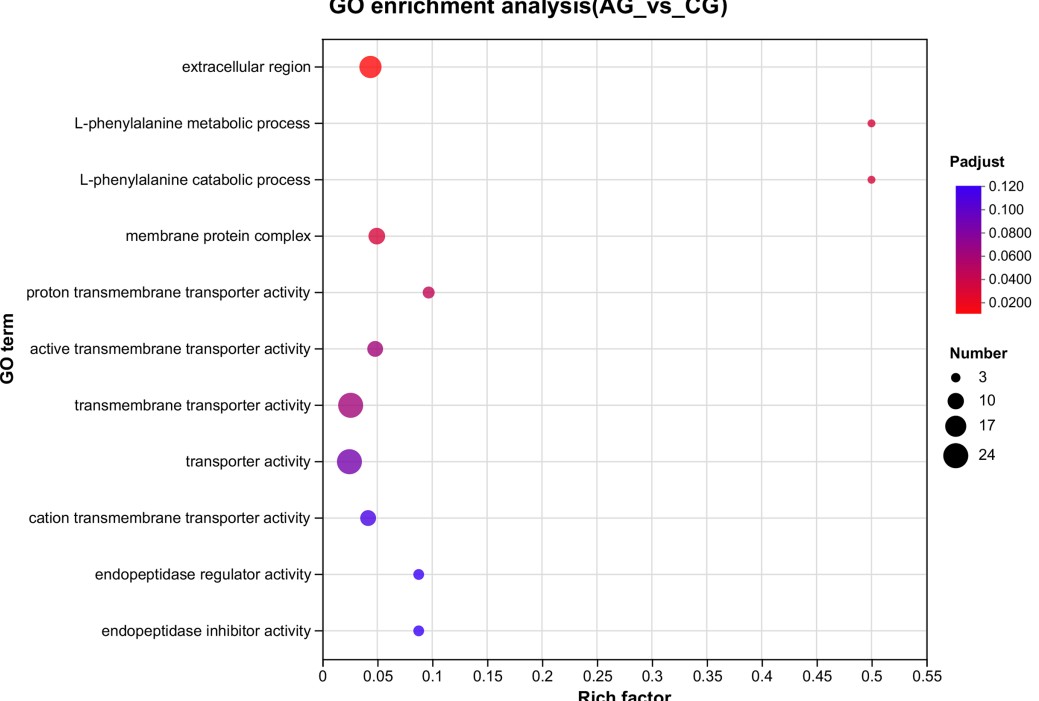

**Figure 3** **The GO enrichments of DEGs under ammonia stress.** The ordinate text indicates the name of the GO, and the description of the classification is as shown in the Class legend information on the right. The abscissa represents Rich factor, that is, the ratio of sample number of enriched genes/transcripts to background number of annotated genes/transcripts in the GO term; The larger the graph, the greater the number of differentially expressed genes. Color represents the significance of enrichment, namely $P_{adjust}$. The darker the color, the more significantly enriched the GO term is, and the color gradient on the right represents the size of $P_{adjust}$.

## Ammonia excretion and ammonia detoxification

Crustaceans have a variety of mechanisms to excrete and detoxify ammonia in a high-ammonia environment (*Zhang et al., 2021*; *Ren et al., 2015*; *Lu et al., 2022*; *Ip, Chew & Randall, 2001*). Crustaceans are ammonia-discharging animals and the gills of aquatic crustaceans are the first point of contact with the outside world and exchange of material (*Henry et al., 2012*). The crustacean gill excretes ammonia in a high ammonia environment through several transporters located in the gill epithelium, such as Rh, NKA, NKCC, and NHE (*Zhang et al., 2021*; *Weihrauch et al., 2002*; *Ren et al., 2015*; *Weihrauch et al., 1999*; *Zhang et al., 2021*; *Martin et al., 2011*). Some studies have reported that ammonia is loaded onto vesicles as $NH_4^+$ and transported along the microtubule network to the apical membrane (*Weihrauch et al., 2002*). Ammonia is released *in vitro* by membrane fusion as well as exocytosis. This ammonia vesicle-trapping mechanism has been demonstrated in *Portunus trituberculatus* and *Carcinus maenas* (*Weihrauch et al., 2002*; *Ren et al., 2015*). In our study, the GO pathways were significantly enriched ($P < 0.05$) and related to ion transport (GO:0006811), inorganic ion transmembrane transport (GO:0098660), anion transmembrane transport (GO:0098656). As expected, most of the DEGs among these three terms were upregulated. We hypothesized that the upregulation of these genes is a protective measure against a high ammonia environment. This result is consistent with

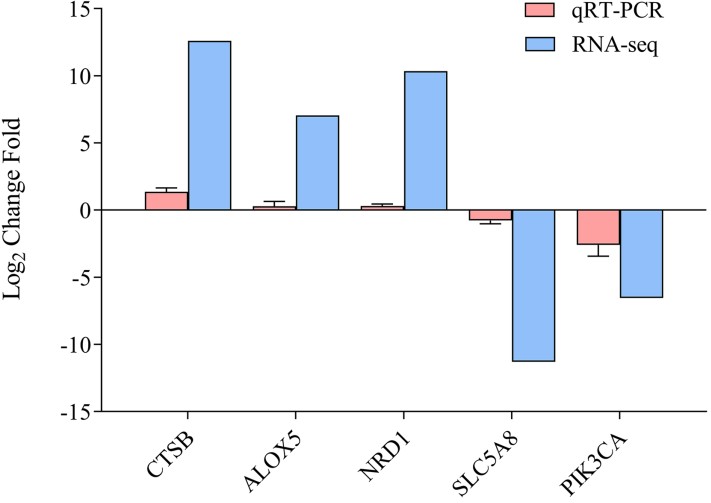

**KEGG enrichment analysis(AG_vs_CG)**

**Figure 4** **Overview of the KEGG pathways significantly enriched in DEGs.** The specific pathways are plotted along the y-axis, and the x-axis indicates the rich factor. The size of the colored dots indicates the number of significantly differentially expressed genes associated with each corresponding pathway: pathways with larger-sized dots contain a higher number of genes. The color of each dot indicates the corrected $P_{adjust}$ for the corresponding pathway.     

**Figure 5** **Comparison of relative fold change of DGEs and qRT-PCR.** Results between ammonia treatment and control group in gills of *E. sinensis*. Gene abbreviations are as follows: *CTSB, Cathepsin B, ALOX5, Arachidonate 5-Lipoxygenase; NRD1, Nardilysin Convertase; SLC5A8, Solute Carrier Family 5 Member 8; PIK3CA, Phosphatidylinositol-4,5-Bisphosphate 3-Kinase Catalytic Subunit Alpha.* Data are presented as mean ± S.D. of three biological replicates (*n* = 3).     

results for *Pacifastacus leniusculus* and *Metacarcinus magister*, in which a high ammonia environment destroyed ion balance and osmoregulation (*Young-Lai, Charmantier-Daures & Charmantier, 1991*; *Harris et al., 2001*).

Crustaceans have an ammonia detoxification mechanism that converts excess ammonia into other compounds, including glutamine and urea, which reduces the concentration of ammonia in the body (*Murray et al., 2003*; *Liu et al., 2014*). Urea levels increase in crustaceans inhabiting high-ammonia environments, including *Marsupenaeus japonicus* and *P. trituberculatus* (*Ren et al., 2015*; *Liu et al., 2014*; *Cheng, Lee & Chen, 2004*). Although urea plays a vital role in the detoxification process of ammonia stress, urea is also toxic and causes adverse effects (*Pan et al., 2018*). Uric acid, which is the precursor to urea, is generated by the breakdown of purine nucleotides, such as adenine and guanine, hypoxanthine, and xanthine. This process is known as the ornithine-urea cycle. Xanthine oxidoreductase, which occurs as two forms of xanthine dehydrogenase (XDH) and xanthine oxidase, plays an important role in the degradation of purine nucleotides. XDH is the rate-limiting enzyme in purine catabolism that catalyzes the conversion of hypoxanthine to xanthine and then xanthine is converted to uric acid (*Pan et al., 2018*; *Lim et al., 2001*). The expression of *XDH* was downregulated, which is notable. Instead, *E. sinensis* utilized metabolic suppression under ammonia stress, which likely lessened the physiological cost of excess urea. *Penaeus monodon* and *P. trituberculatus* both showed similar results (*Pan et al., 2018*; *Li et al., 2018*).

The breakdown of proteins into amino acids releases ammonia, and may decrease protein and amino acid hydrolysis to avoid ammonia accumulation in the body (*Ren & Pan, 2014*). Our previous study reported that protein degradation was inhibited in the hepatopancreas of *E. sinensis* under the same ammonia concentration. Inhibiting the degradation of protein would reduce endogenous ammonia accumulation, which agrees with previous studies (*Tang et al., 2022*; *Lim et al., 2001*; *Ren & Pan, 2014*; *Saha, Dutta & Bhattacharjee, 2002*; *Li & Xiang, 2013*).

## Immune response

*E. sinensis* lacks acquired immunity, so it must rely on the innate immune system, which is comprised of cellular and humoral immunity (*Jia et al., 2017*). In crustaceans, hemolymph is important in the non-specific immune defense system. Previous research has shown that the total haemocyte count (THC) of crustaceans, such as *E. sinensis* and *Litopenaeus vannamei*, is lower under ammonia stress than that under normal conditions. The decrease in THC demonstrated that ammonia stress damages immunocompetence (*Hong et al., 2007b*; *Luzio, Pryor & Bright, 2007*). Phagocytosis eliminates metabolic waste, apoptotic cells, and external pathogens that infiltrate the body and is the most crucial defense mechanism of hemolymph. Lysosomes are active organelles that take inputs from membrane trafficking *via* secretion, endocytosis, autophagy, and phagocytosis (*Luzio, Pryor & Bright, 2007*). In our study, the GO analysis revealed that the DEGs related to membrane protein complex (GO:0098796), endopeptidase regulator activity (GO:0061135), endopeptidase inhibitor activity (GO:0004866) and peptidase inhibitor activity (GO:0030414) varied significantly. The KEGG analysis showed that the DEGs related to

phagosome (ko04145), lysosome (ko04142) and the T cell receptor signaling pathway (ko04660) varied significantly, which explains how ammonia inhibited the immune response (Figs. 3 and 4).

Furthermore, we identified *UCHL3* and *OGT* related to immunity, which were differentially expressed. Ubiquitination is an important post-translational modification of eukaryotic proteins and is involved in most signaling pathways. Ubiquitination is reversible, and deubiquitination plays a crucial role in many functions, such as apoptosis, tumor growth, cell cycle, and antioxidant immunity (*Mevissen & Komander, 2017*). Ubiquitin C-terminal hydrolases (UCHs) are a subfamily of deubiquitinating enzymes that reverse the protein ubiquitination process (*Zeichner, Terawaki & Gogineni, 2016*). UCHL3 also participates in immune regulation. A previous study reported that UCHL3 in *Macrobrachium nipponense* is involved in innate immunity to resist bacterial invasion by activating the NF-κB signaling cascade (*Zhu et al., 2022*; *Cheng et al., 2021*). Additionally, another key to post-translational modification of proteins is O-glycosylation. Previous studies have revealed that proteins with a single N-acetylglucosamine fragment added through O-glycosylation are related to the immune response, including heat shock factor 1 and heat shock protein (*Chien et al., 2020*; *Xiao et al., 2019*). Upregulating *ogt* would code more protein O-glycosylation in the hepatopancreas and hemocytes of *L. vannamei*, which would increase immune capability (*Cheng et al., 2021*). The expression of these two genes was downregulated, suggesting the immune disorder in *E. sinensis* under ammonia stress in our study.

The same conclusion was drawn from the hepatopancreas of *E. sinensis* under the same stress duration and concentration. The hepatopancreas is an important detoxification organ that contains a large number of immune-related genes and pathways. The genes in the hepatopancreas were downregulated (*Tang et al., 2022*). The gill and hepatopancreas results indicated that high ammonia exposure may have damaged the immune system of *E. sinensis*, which agrees with results on *L. vannamei*, *P. monodon*, and *Procambarus clarkii* (*Li et al., 2018*; *Xiao et al., 2019*; *Luo et al., 2022*).

## Apoptosis

Apoptosis is activated by ammonia in crustaceans. Apoptosis is not an autogenous injury phenomenon under pathological conditions, but a death process to better adapt to the living environment (*Fromentel & Soussi, 1992*). The apoptosis pathway was changed significantly in our study ($P < 0.01$). Additionally, the regulation of the genes encoding the apoptosis-related enzymes cathepsin B (CTSB) and ubiquitin conjugating enzyme E2 W (UBE2W) proteins also changed significantly. The cysteine family includes the *CTSB* gene, which is frequently involved in apoptosis. According to previous studies, *L. vannamei*, *Fenneropenaeus chinensis*, and *Palaemonetes pugio* are protected from external stress by apoptosis caused by the upregulation of *CTSB* (*Guo et al., 2013*; *Griffitt et al., 2006*). Downregulation of *UBE2W* in mouse spermatogenic cells increases the expression of P53, caspase 6, and caspase 9, which induces apoptosis. The expression of *Bcl-2* decreased significantly, which inhibited cell apoptosis. Downregulation of *UBE2W* activates the P53/Bcl-2/caspase 6/caspase 9 signaling pathway, which increases the rate of cell apoptosis (*Lei*

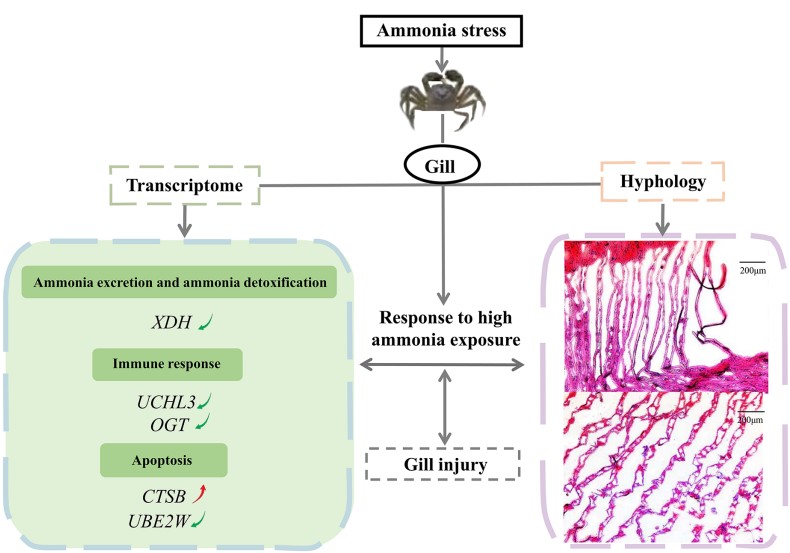

**Figure 6 The molecular mechanism of *E. sinensis* gills response to ammonia exposure.**

*et al., 2020*). The upregulation of *CTSB* and downregulation of *UBE2W* further confirmed that ammonia stress might activate apoptosis in *E. sinensis*. This result explains the injury to the gill under ammonia stress at the molecular level.

# CONCLUSIONS

In conclusion, the study investigated the effects of a high ammonia concentration on the gill mechanism of *E. sinensis* at the transcriptome level. We demonstrated the molecular gill mechanism in *E. sinensis* during the response to high ammonia exposure (Fig. 6). In this study, gills under ammonia stress were damaged at the histological level. The transcriptome analysis identified many DEGs related to ammonia excretion and detoxification, the immune response, and apoptosis, but most were downregulated. Taken together, we inferred that *E. sinensis* exhibited a reduced immune response due to high ammonia toxicity. This study revealed the molecular mechanism of *E. sinensis* gills in response to ammonia stress at the transcriptional and histological levels and provides insight for further study of ammonia stress in crustaceans.

# ACKNOWLEDGEMENTS

Thanks to Ms. Shang Zhu for the help in sample collection.

## Funding

This study was funded by the National Natural Science Foundation of China (grant number 32370436), the Education and Teaching Reform Program of Yancheng Teachers University (2021YCTCJGZ002), and the Open Foundation of Jiangsu Key Laboratory for Bioresources of Saline Soils (grant number JKLBS2019006). The funders had no role in

study design, data collection and analysis, decision to publish, or preparation of the manuscript.

## Grant Disclosures

The following grant information was disclosed by the authors:
National Natural Science Foundation of China: 32370436.
Education and Teaching Reform Program of Yancheng Teachers University: 2021YCTCJGZ002.
Open Foundation of Jiangsu Key Laboratory for Bioresources of Saline Soils: JKLBS2019006.

## Competing Interests

The authors declare that they have no competing interests.

## Author Contributions

- Zhengfei Wang conceived and designed the experiments, prepared figures and/or tables, authored or reviewed drafts of the article, and approved the final draft.
- Yue Wang performed the experiments, analyzed the data, prepared figures and/or tables, authored or reviewed drafts of the article, and approved the final draft.
- Yayun Guan analyzed the data, authored or reviewed drafts of the article, and approved the final draft.
- Zhuofan Chen analyzed the data, prepared figures and/or tables, authored or reviewed drafts of the article, and approved the final draft.
- Yaotong Zhai performed the experiments, authored or reviewed drafts of the article, and approved the final draft.
- Ya Wu analyzed the data, authored or reviewed drafts of the article, and approved the final draft.
- Ying Zhou analyzed the data, authored or reviewed drafts of the article, and approved the final draft.
- Jinghao Hu analyzed the data, authored or reviewed drafts of the article, and approved the final draft.
- Lulu Chen conceived and designed the experiments, prepared figures and/or tables, authored or reviewed drafts of the article, and approved the final draft.

## DNA Deposition

The following information was supplied regarding the deposition of DNA sequences:
The RNA databases are available at GenBank: SRR22094391, SRR22094390, SRR22094389, SRR22094388, SRR22094387.

## Data Availability

The raw data used for qRT-PCR analysis is available in the Supplemental File.

## Supplemental Information

Supplemental information for this article can be found online at http://dx.doi.org/10.7717/peerj.16786#supplemental-information.

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
