# Peer review of "Transcriptome analysis of Chinese mitten crabs (Eriocheir sinensis) gills in response to ammonia stress"

_PeerJ, doi:10.7717/peerj.16786_

## Round 0.1 · original submission · Major Revisions

The manuscript by Wang et al. needs strong revision. The objectives of the manuscript are not clear. The methodology is ambiguous and the following points need clarification.

1. Why De novo assembly has been used if the reference-based assembly is available? I am interested to look for comparative data of both assemblies.

2.  In experiments authors use a concentration of ammonia as per Hong etc 2007. However, The experiments need to be supplemented for the toxicity data of ammonia under various concentrations and time points to justify the ideal conc. of 325.07 mg/L used in the study.

Reviewer 1 ·

Basic reporting

The article is written in clear, unambiguous, professional English.

The structure and flow of the article is good.

The RNA-seq dataset generated is publicly available at NCBI SRA.

Figures are of good quality.

Experimental design

The research question is well defined and experiment design is appropriate.

Details of some of the bioinformatics analysis are missing in the description of the methods, which I touch on below. In the interest of openness and reproducibility, I suggest the authors to either make the code they used for analysis publicly available or included in the Supplementary Material.

Another major concern is the very small sample size (only 2 individuals in the experiment group) which I comment on below.

Validity of the findings

Despite some issues with their methods, the results seem largely valid, and the discussion and conclusion sections are well written.

Additional comments

Following are some comments that I think would help the manuscript and address issues of scientific reproducibility.

- There are only 2 individuals in the experimental group. The authors should discuss how this might have affected the results of analysis, the interpretation of those results, and ultimately the generalizability of their results.

- The section describing RNA extraction and sequencing ( Line 180~) should make reference to Table 2. Table 2 should contain a column with the NBCI SRA accession IDs. The number of reads in Table 2 do not match the number of reads in NBCI SRA. Did the authors submit only a part of their data?

- In Line 113,114: ".. biological and technical replicates ..." --> there are no technical replicates in this study.

- Details of some important bioinformatics analysis is missing.
-- The authors do not describe how reads per kilobase per million was computed.
-- Parameters used for annotation is not described. In Table S2, what is the meaning of the third column?
-- edgeR expect actual/raw read counts, and normalized counts like rpkm, but the authors claim to have normalized it before DEG. This might be producing wrong results.

- Related to the comment above, I suggest the authors to either make the code they used for analysis publicly available or included in the Supplementary Material.

- Line 233: The authors claim that the RNA-seq and qRT-PCR results are consistent even though Figure~5 suggests that assuming qRT-PCR is correct RNA-seq is over-estimating both over-expression and under-expression by at times 1000 fold.

Reviewer 2 ·

Basic reporting

Dear Editor,

I have finished reviewing the article titled "Transcriptome analysis of Chinese mitten crabs (Eriocheir sinensis) gills in response to ammonia stress." This study delved into the ammonia detoxification and extracellular release of nitrogenous compounds in gill tissue at a transcriptome level. The research was conducted on a crab species that can survive and thrive in rice fields and is exposed to high levels of nitrogen fertilizer or ammonia. In this context, the study was found exciting and suitable for evaluation. However, the study needs to include a more methodological axis, and publishing it in the current situation is not appropriate.I have provided clarifications regarding the limitations of the study mentioned below.

Experimental design

i) Even though the crab species chosen for study (Eriocheir sinensis) already has a reference genome and annotation from 2022 (see: https://www.ncbi.nlm.nih.gov/datasets/taxonomy/95602/), why was a de novo transcriptome analysis conducted? De novo transcriptome analysis is appropriate and necessary for organisms that lack a reference genome, but it comes with significant issues, such as the creation of chimeric transcripts and the inability to differentiate between isoforms and genes in gene expression analysis. As a result, a reference-based transcriptome analysis approach must be utilized to conduct the analysis. However, the de novo transcriptome approach can still be employed to discover novel transcripts, as long as the novel transcript obtained aligns with the reference genome in a "splice-aware" manner.

ii) In the "Analysis of differentially expressed genes (DEGs)" section, the phrase "The expression levels of the unigenes were calculated by reads per kilobase per million" is mentioned. However, the specific method and software used for RPKM calculation are not specified.

iii) The de novo assembly analysis mentions that the CG-3 sample was removed due to its poor quality and there were only two sets of CG available for supplement in this season. However, when examining Table 2, it seems that the quality of the CG-3 sample was actually acceptable.

iv) It is unclear whether the differential expression analysis is being conducted at the isoform level or the gene level, as this information has not been disclosed.

v) When conducting de novo transcriptome analysis on environmental samples, it is common to observe bacterial or protist contamination. During RNA isolation and sequencing, these contaminants are included in the analysis. The question is whether or not the de novo transcriptome analysis successfully removed these contaminants.

vi) In addition, a PCA analysis should be performed to differentiate the samples between the control and experimental groups regarding gene expression level.

Validity of the findings

The analysis approach currently used may result in findings of weak validity.

Additional comments

I strongly recommend making reference-based transcriptome analysis instead of de novo, which paves the way for many false-positive results.

---

## Round 0.2 · Minor Revisions

The manuscript contains numerous bioinformatics concerns to be considered for publication. I am interested to see reference based results as supplementary data file. Authors should clearly mention in the manuscript the difference between reference based and de-novo results.

Reviewer 1 ·

Basic reporting

no comment

Experimental design

no comment

Validity of the findings

no comment

Additional comments

In my last review, I pointed out some issues in the paper, many of which concerned technical details about the bioinformatics procedures.
The authors have not responded adequately to some of my concerns which I re-iterate here.


1. There are only 2 individuals in the experimental group. The authors should discuss how this might have affected the results of analysis, the interpretation of those results, and ultimately the generalizability of their results.

In their response, the authors describe outlier removal, which is not the point of this question.

2.The authors do not describe how reads per kilobase per million was computed.

In their response, the authors defined RPKM and mentioned that edgeR was used for "estimate recounts". They have not responded to the question about how counts were obtained.

3.edgeR expect actual/raw read counts, and normalized counts like rpkm, but the authors claim to have normalized it before DEG. This might be producing wrong results.

In their response, the authors mention normalizing by gene length, which is not required when comparing the same gene across samples/conditions.

The following is taken from Section 2.8.1 of edgeR user guide on Bioconductor (https://www.bioconductor.org/packages/devel/bioc/vignettes/edgeR/inst/doc/edgeRUsersGuide.pdf)

"... For example, read counts can generally be expected to be proportional to length as well as to expression for any transcript, but edgeR does not generally need to adjust for gene length because gene length has the same relative influence on the read counts for each RNA sample. For this reason, normalization issues arise only to the extent that technical factors have sample-specific effects ..."


4. Given that some key information about the data analysis procedure are missing, I strongly recommend, for the sake of reproducibility, that the authors either make the code they used for analysis publicly available or included as supplementary material.

---

## Round 0.3 · Major Revisions

The authors should address the comments of reviewer 2, especially focusing on reference genome-based studies.

Reviewer 1 ·

Basic reporting

no comment

Experimental design

no comment

Validity of the findings

no comment

Reviewer 2 ·

Basic reporting

After carefully analyzing the edits plus improvements made to the article titled "Transcriptome analysis of Chinese mitten crabs (Eriocheir sinensis) gills in response to ammonia stress," I have identified some deficiencies that still need to be addressed.

1) To effectively address the criticisms and suggestions provided by reviewers regarding the manuscript, the authors should respond item by item to each of them. The "response-to-reviewers" document must include the comments written by the referees verbatim and provide their responses underneath. However, upon reviewing the "response-to-reviewers" document, I noticed none of my sentences were included. It makes the review process for the article significantly more difficult.

2) Upon examining the "response-to-reviewers" file, some of my comments were barely mentioned. For instance;
"iv) It is unclear whether the differential expression analysis is being conducted at the isoform level or the gene level, as this information has not been disclosed.

v) When conducting de novo transcriptome analysis on environmental samples, it is common to observe bacterial or protist contamination. During RNA isolation and sequencing, these contaminants are included in the analysis. The question is whether or not the de novo transcriptome analysis successfully removed these contaminants."

3) In my reviewer report, I recommend that because the crab species subject to the study has a reference genome and annotation (https://www.ncbi.nlm.nih.gov/datasets/taxonomy/95602/), there is no need to perform a de novo transcriptome assembly analysis. De novo transcriptome assembly must only be performed if the reference genome is not available because there are some drawbacks in de novo transcriptome analyses, such as the creation of chimeric transcripts and the inability to differentiate between isoforms and genes in gene expression analysis. I mentioned that if a study is designed only on novel transcripts, de novo transcriptome analysis can be used. Excluding de novo transcriptome assembly analysis, this study's RNA-Seq setup is based on cross-group DEG analysis using only reference-based transcriptome analysis.

4) Reference-based RNA-Seq analysis involves the use of Cufflinks and RPKM methods. However, these programs are now outdated, having been developed almost a decade ago. Furthermore, the manuscript does not mention which alignment program is being used for the analysis. To ensure the accuracy of the results, it is recommended to use up-to-date alignment programs like Hisat2 or STAR. For abundance estimation analysis, it is also advisable to make use of state-of-the-art software such as Deseq2, edgeR, or limma with the most recent version.

Experimental design

a) must performe reference-based transcriptome analysis,
b) use most recent splice-aware aligners and DEG analysis methods,

Validity of the findings

The analysis approach currently used may result in findings of weak validity.

Additional comments

I strongly recommend making reference-based transcriptome analysis instead of de novo, which paves the way for many false-positive results.

---

## Round 0.4 · accepted · Accept

The manuscript has been revised fully and there is no more concern. I feel pleased to accept the manuscript for publication in PeerJ

Reviewer 2 ·

Basic reporting

The author response to my recommended revisions is largely safisfactory. I would like to remind the authors that protists are eukaryotic organisms.

Experimental design

no comment

Validity of the findings

no comment

Additional comments

no comment